# Control of filament length by a depolymerizing gradient

**Arnab Datta** *, **David Harbage**, **Jane Kondev**

Department of Physics, Brandeis University, Waltham, Massachusetts, United States of America

* adatta@brandeis.edu

**Data Availability Statement:** All relevant data are within the manuscript.

**Funding:** This work was supported by the National Science Foundation grants DMR-1610737(JK) and MRSEC-1420382(JK), and by the Simons Foundation (www.simonsfoundation.org/)(JK). The

## Abstract

Cells assemble microns-long filamentous structures from protein monomers that are nanometers in size. These structures are often highly dynamic, yet in order for them to function properly, cells maintain them at a precise length. Here we investigate length-dependent depolymerization as a mechanism of length control. This mechanism has been recently proposed for flagellar length control in the single cell organisms Chlamydomonas and Giardia. Length dependent depolymerization can arise from a concentration gradient of a depolymerizing protein, such as kinesin-13 in Giardia, along the length of the flagellum. Two possible scenarios are considered: a linear and an exponential gradient of depolymerizing proteins. We compute analytically the probability distributions of filament lengths for both scenarios and show how these distributions are controlled by key biochemical parameters through a dimensionless number that we identify. In Chlamydomonas cells, the assembly dynamics of its two flagella are coupled via a shared pool of molecular components that are in limited supply, and so we investigate the effect of a limiting monomer pool on the length distributions. Finally, we compare our calculations to experiments. While the computed mean lengths are consistent with observations, the noise is two orders of magnitude smaller than the observed length fluctuations.

## Author summary

A fundamental problem in cell biology is how molecules self-organize into functional structures. Flagella and cilia are filamentous structures made of hundreds of thousands of proteins, which undergo rapid turnover, and for them to function properly cells need to maintain them at a precise length. Based on experiments on flagellated single cell organisms, we consider a model of length control by a depolymerizing gradient. Such gradients form when depolymerizing proteins are transported to the tip of the filament by a combination of diffusion and directed transport. The result is an accumulation of depolymerizing proteins at the tip of the filament leading to length-dependent depolymerization of the filament. This negative feedback of length on filament assembly provides length control for multiple filaments even when all the molecular components are drawn from a shared cytoplasmic pool. Using a chemical master equation description of this model of length control, we compute the distribution of filament lengths, and analyze their dependence

funders had no role in study design, data collection and analysis, decision to publish, or preparation of the manuscript.

**Competing interests:** The authors have declared that no competing interests exist.

on model parameters. We find good agreement between our model and experiments, and suggest new experiments to test model predictions.

## Introduction

All eukaryotic cells make nanometer-sized proteins that polymerize to form filaments, which are hundreds of nanometers to tens of microns in length. These filaments often associate with other proteins to form larger structures that perform critical roles in cell division, cell motility, and intracellular transport. They are often highly dynamic, experiencing a high turnover rate of their constitutive protein subunits [1,2]. Despite this, in order to function properly some of these structures, like the mitotic spindle, or actin cables in budding yeast [2,3], must maintain specific and well-defined sizes.

When considering dynamics of filament length, the two basic processes are the addition of monomer units, characterized by the rate of assembly, and the removal of these units, which is described by the rate of disassembly. For the dynamics to reach a steady state with a well-defined length, one or both rates must be length dependent. Therefore, the search for the molecular mechanism of length regulation often boils down to finding out whether the rate of assembly or disassembly (or both) is length dependent, and what molecular-scale interactions produce this length dependence [4,5].

Many mechanisms for length regulation have been proposed based on careful experiments in cells and on reconstituted filamentous structures in vitro [3,6–9]. One well studied mechanism of length control is to limit the number of subunits available for assembly [10]. For example, in vitro experiments have demonstrated how the size of the mitotic spindle depends on the size of the compartment in which it self-assembles, consistent with a limiting-pool mechanism of length control [11]. In the case of multiple structures, all assembled from the same pool of components, this mechanism can only control the total number of subunits in all the structures, but is incapable of controlling the size of each individual structure [12]. How cells create and maintain the lengths of multiple filamentous structures, such as cilia in a multiciliate cell, is the key question that motivates this paper.

The question of length control of multiple filaments that share a monomer pool has been studied extensively in the single cell algae Chlamydomonas reinhardtii ("Chlamy") [13–15]. There, experiments have directly demonstrated that the assembly dynamics of the two flagella are coupled. If one flagellum is cut, the intact flagellum shrinks while the cut flagellum grows, until the two reach the same length, which is smaller than the length before the cut. Once the lengths of the two flagella equalize, they continue growing at the same rate until the original lengths are reached, and the lengths stabilize. In experiments where drugs are added that prevent protein production, after both flagella are cut, they regrow to a length that is shorter than their original length. These experiments indicate that the free tubulin pool, or some other protein required for assembling a flagellum, is limiting, and is shared between the two flagella [15]. Furthermore, these experiments imply the presence of a length control mechanism that is able to detect the lengths of individual flagella, which the simple limiting-pool mechanism cannot do as it is only sensitive to the total number of monomers in both flagella [12].

The mechanism of length control of the two Chlamy flagella that has thus far received the most attention is one that focuses on length dependent assembly [16,17]. Recently it has been shown that length dependent assembly can arise from a limited and shared pool of motors that transport tubulin to the flagellar tip, where assembly occurs [18]. In this case, though, the assembly rate for the two flagella is the same and depends on the sum of their lengths.

Therefore, this mechanism still fails to control individual filament lengths and it has been proposed instead that length dependent depolymerization might be the key length controlling process. Moreover, in a different single cell organism, Giardia lamblia ("Giardia"), which has four pairs of flagella of different length, recent experiments have shown that the assembly rate is independent of flagellar length, also suggesting length-dependent depolymerization as the mechanism of length control [19].

Examples of length-dependent depolymerization are abundant in the literature. In vitro studies of kinesin-8 have directly demonstrated that its rate of depolymerization scales linearly with the length of the microtubules it binds to [20]. Other, in vivo studies have demonstrated that the microtubule depolymerizing kinesin-13 localizes to the tips of flagella, and that reduction of its expression in the cell leads to longer flagella [21–23]. In Giardia this localization was shown to be quantitatively consistent with directed transport of kinesin-13 to the tip of the flagellum, which, in combination with the depolymerizing activity of this non-motile kinesin, leads to a length dependent depolymerization [19].

In this paper we examine how a gradient of a depolymerizing protein along the length of a filament can control the length of multiple filaments via length dependent depolymerization. Using a master equation for the lengths, we show that length dependent depolymerization can produce well defined lengths for multiple filaments, even when the dynamics of individual filaments are coupled by a limiting pool of monomers. For parameter values typical of a Chlamydomonas cell, our model produces steady state filament lengths comparable to the lengths of flagella in these cells, and predicts steady state fluctuations in length much smaller than the average.

## Results

### Depolymerization gradient

Length dependent depolymerization has been shown to occur when a depolymerizing factor is actively transported to the ends of filaments by the action of motors [3,19,20]. The length dependence of the depolymerization rate arises due to the formation of a spatial gradient of the depolymerizing factor along the length of the filament (see Fig 1). Quantitative measurements using fluorescently labeled proteins have uncovered gradients that vary linearly [20] and exponentially [19,24] with distance along the length of the filament. Here we consider both types of concentration gradients of depolymerizing proteins and how they result in length control of cytoskeleton filaments.

Linear and exponential gradients arise from combined directed transport and diffusion of molecules within a flagellum, as shown in Fig 1. In both cases the molecule (depolymerizing protein) is transported by motors toward the tip of the flagellum, where it is released. In the case when the loading of the depolymerizer only occurs at the proximal end of the flagellum, a linear gradient is obtained:

$$C(x) = C_0 \left( 1 + \frac{x}{\lambda_\ell} \right). \tag{1}$$

Here, $C_0$ is the concentration of depolymerizers in the shared (cytoplasmic) pool and $\lambda_\ell$ is the length scale of the gradient. A linear gradient was observed directly in experiments with fluorescently labeled kinesin-2 in Chlamy. These kinesins are loaded at the proximal end of the flagellum, move directionally toward the distal end, and return back to the base diffusively [25], but have no known depolymerizing activity. Whether other Chlamy proteins, localized to the flagella, exhibit linear concentration profiles, is not known.

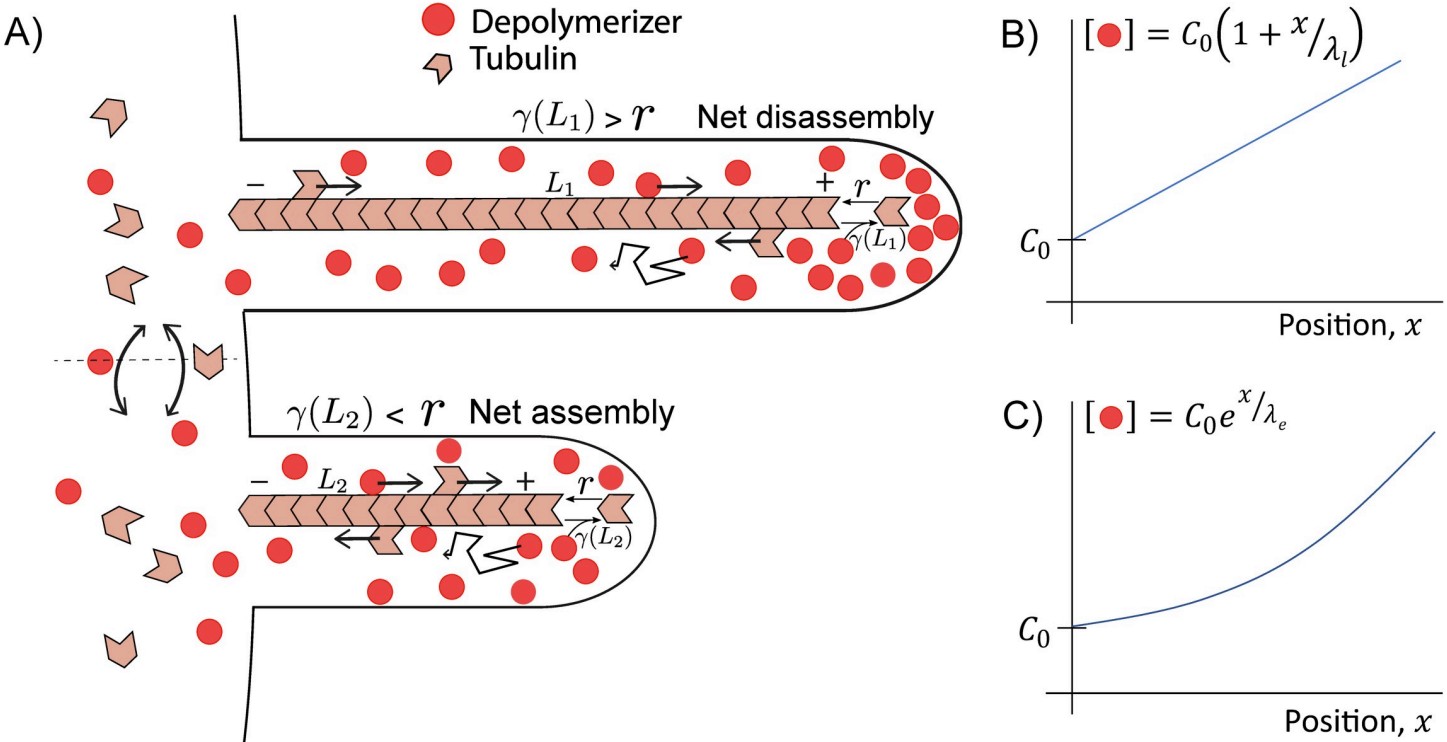

**Fig 1. Schematic of a depolymerizing gradient. A.** Depolymerizing proteins (red) accumulate at the tip of filament by directed transport from – to + end, and diffusion from the tip (+ end), where they are released. The resulting length dependent concentration of depolymerizing proteins at the filament tip leads to a length dependent depolymerization rate $\gamma(L)$. Assembly also occurs at the tip at rate ($r$), which is determined by the monomer (tubulin) pool in the cell body. Flagella draw all protein components from a common pool. **B.** A linear concentration gradient forms when the depolymerizing proteins can only attach to the motors (not shown) at the base (– end) of the filaments. **C.** An exponential concentration gradient forms when the depolymerizers can bind to motors anywhere along the filament. $C_0$ is the concentration of depolymerizers in the cytoplasm, and $\lambda$ is the characteristic length scale of the gradient.

In the second case that we consider, the depolymerizer is free to bind to the motors that are moving along the filament anywhere in the flagellum. (Note that whether the depolymerizer binds to a motor directly, or to other cargo proteins carried by the motor, does not affect our results.) The concentration gradient of the depolymerizers, in this case is given by,

$$C(x) = C_0 e^{x/\lambda_e}. \tag{2}$$

Once again, $C_0$ is the concentration of depolymerizers in the shared pool and $\lambda_e = \sqrt{D/k_{Bind}}$ is the length scale of the gradient. $D$ is the diffusion coefficient of the depolymerizers in the flagellum and $k_{Bind}$ is the rate with which they bind to the motors moving along the filament [19,24].

The presence of a concentration gradient of a depolymerizing protein leads to length dependent depolymerization. Namely, we assume that disassembly of the filaments only happens at the tip of the filament, $x = L$, as is observed in Chlamy [26–29]. For the depolymerizer to remove a monomer from the tip of the filament, it must first diffusively find the end of the filament, bind to it, and then remove a subunit. We assume that depolymerizer concentration is low, so that binding to the end of the filament is the rate limiting step. In that case, the rate of depolymerization is proportional to the concentration of the depolymerizer in the vicinity

of the tip,

$$\gamma(L) = k_- C(L). \tag{3}$$

For example, in vitro studies of MCAK, a depolymerizing non-motile kinesin, show a linear relation between the depolymerization rate and MCAK concentration, for concentrations less than 10 nM [22]. While there is no evidence for MCAK being the depolymerizer that controls flagellar length, we mention it here as an example of the mechanism of action by a depolymerizing protein on the assembly dynamics of microtubules, which we adopt in our model. Specifically, it is important to distinguish this mechanism, which depends on the freely diffusing concentration of depolymerizer in the vicinity of the filament tip, from the action of kinesin-8 that walks along the filament toward the tip, and knocks off monomers as it unbinds from the tip[20].

Our model of depolymerization does not take into account the stochastic dynamics of the depolymerizing protein, and instead treats it as being in steady state as the length of the filament changes. This is justified by assuming that the timescale associated with the relaxation of the gradient, i.e., the time to reestablish the gradient after a change in filament length has occurred, is fast compared to the timescale associated with adding or removing a filament subunit. This separation of timescales was observed experimentally in Giardia cells [19]. In Giardia, the rate of flagellar growth is about a micron per hour, after the cells have been treated with Taxol. The depolymerizing protein was identified as kinesin-13, which was shown to exhibit an exponential gradient along the length of the filament. When one micron of the filament was photobleached from the tip, it was found that the timescale of recovery is about one min, much faster than the time for the flagellum to extend by one micron. Similar observations were made regarding the linear gradient of kinesin-2 along the flagella in Chlamy, where the recovery after photobleaching occurred in three seconds [25]. In these cells the rate of filament growth is of order microns per minute, i.e., it unfolds on a considerably slower timescale. While kinesin-2 is not a depolymerizing protein, we expect similar dynamics of other proteins that are carried by intraflagellar transport as cargo. Based on these observations, we treat the gradient dynamics in an adiabatic approximation where they relax instantaneously to the new filament length as the filament undergoes its stochastic dynamics, as implied by Eq 3.

The polymerization rate we take to be proportional to the concentration of free monomers in the shared cytoplasmic pool, which bind to motors in the flagellum and are transported by them to the tip where assembly occurs (Fig 1A):

$$r(\{L_i\}) = k_+ (N - \sum_{i=1}^{M} L_i). \tag{4}$$

Here $N$ is the total number of monomers, $M$ is the number of filaments, and $L_i$ is the length of the $i^{th}$ filament in monomer units (for example, one micron of a microtubule corresponds to roughly 2000 tubulin proteins [30]). We assume that the assembly rate per monomer ($k_+$) is length independent, so that length dependence of assembly can only come from the pool of monomers being significantly depleted by flagellar assembly. In particular, we do not take into account the length dependence of $k_+$ that can arise due to a limited pool of motors that transport monomers to the tip of the flagellum (Fig 1), which was the case analyzed in ref.[18]. There it was shown that, while this feature of the assembly dynamics is important for reproducing the observed dynamics of Chlamy flagella right after one of them is cut, it alone does not provide length control for multiple flagella. Moreover, experiments on flagella in Giardia cells [19] found no such length dependence of $k_+$, which is another reason we choose to leave it out of our analysis, and focus solely on length dependent depolymerization.

Given that the process of polymerization encompasses many molecular steps, describing it with a single rate constant $k_+$ might seem like an oversimplification. Still, we believe it is a reasonable assumption given that the timescale of consecutive arrivals of IFT trains that carry tubulin are of order seconds, both in Chlamy and in Giardia [19]. Given the growth rate of microns per minute in Chlamy (and microns per hour in Giardia), this implies a few hundred tubulin dimers arriving with each train, given the estimate of about 20,000 tubulin dimers per micron of flagellum (see Methods). In other words, the timescales at which individual building blocks (tubulin dimers) arrive is very small compared to the timescales of length fluctuations, which justifies using a constant assembly rate per monomer, $k_+$.

In Chlamy, experiments suggest that a substantial fraction of the cell's tubulin is used up by its two flagella, and the presence of other microtubule structures within the cell body further decrease the amount of free tubulin [15,31,32]. In Giardia, the tubulin pool seems to be much larger as there is no evidence for its depletion [19], consistent with the observation that the polymerization rate is length independent. Below, we consider both the case of a finite and an effectively infinite pool of shared monomers, and show that length dependent depolymerization leads to filaments with well-defined lengths in both situations.

The model we adopt for the dynamics of flagella describes each flagellum as a single filament and does not take into consideration the fact that the flagellum is composed of multiple microtubule doublets and other structural components [16]; we take this structure into account only in the sense that each monomer that attaches or detaches to the end of the filaments contributes to a change in length that is only a small fraction of the monomer size. Specifically, we estimate that a micron of the flagellum is made of 20,000 tubulin proteins (see Methods).

## Length control by length dependent depolymerization

We use the length-dependent disassembly model described above to compute the mean filament length and fluctuations around the mean, in steady state. The steady state length is obtained by finding the filament length for which the rate of polymerization and depolymerization are equal. To compute the length fluctuations, we make use of the chemical master equation for the probability distribution of filament lengths, which we solve in steady state.

**A. Large monomer pool.**  First, we consider the case when the monomer pool is large enough so that in steady state the monomers taken up by the filaments are only a small fraction of the total number of monomers ($N$) in the cell, $\Sigma_i L_i \ll N$. In this case we can approximate the assembly rate as a constant, independent of length $r \approx k_+ N$; $k_+$ is the rate of assembly per monomer. Recent experiments in Giardia found a length independent polymerization rate for its flagella, consistent with this assumption. Furthermore, in this case the dynamics of individual filaments are uncoupled, and we can restrict our analysis to an individual filament.

In steady state, the rate of polymerization and depolymerization are equal. Using Eq(3) for the disassembly rate and $r \approx k_+ N$ for the assembly rate, we find the steady state length

$$L_{ss} = \lambda_l (QN - 1), \tag{5}$$

where we have used Eq (1) for the concentration of the depolymerizer at the plus end ($x = L$) of the filament. Here we have introduced a dimensionless quantity $QN = \frac{k_+ N}{k_- C_0}$. This quantity is the ratio of the rate of polymerization and the minimum rate of depolymerization, when the concentration of depolymerizing proteins is equal to the cytoplasmic concentration $C_0$. As will become clear later, this is the key dimensionless quantity that controls both the steady state filament length and the length fluctuations.

Similarly, for the case of the exponential gradient of depolymerizers, using Eq (2), we find

$$L_{ss} = \lambda_e \, log(QN). \tag{6}$$

Comparing Eqs (5) and (6) we see a clear difference in the way in which the linear and exponential gradients of depolymerizers determine the steady state filament length. In the case of the exponential gradient, Eq (6), the filament length is set solely by the length scale of the gradient $\lambda_e$, since for reasonable values of the parameters (discussed below), $log(QN)$ is of order one. This is unlike the case of the linear gradient, where the filament length is proportional to $QN$, and therefore the steady state length is much more sensitive to the chemical rate constants, as well as the amount of depolymerizers ($C_0$) and monomers ($N$) in the cell.

To quantify the precision of this length control mechanism, we compute the fluctuations in length around the steady state value using the chemical master equation

$$\frac{dP(L,t)}{dt} = r \, P(L-1,t) + \gamma(L+1)P(L+1,t) - P(L,t)(r + \gamma(\mathrm{L})), \tag{7}$$

where $P(L,t)$ is the probability that at time $t$ a filament has length $L$. Once again, the length dependent depolymerization rate $\gamma(L)$ is given by Eq (3), while the polymerization rate is constant and given by $r = k_+N$.

For the linear depolymerizer gradient we find the steady state probability distribution,

$$P_l(L) = \frac{1}{Z}\frac{(QN\lambda_\ell)^L}{\Gamma(L + \lambda_\ell + 1)}. \tag{8}$$

where $1/Z$ is the normalization constant and $\Gamma$ is the gamma function. The approximate mean of the distribution is $\langle \mathrm{L}\rangle_l \approx (QN-1)\lambda_l$, which is also the steady state length; see Eq (5). The variance of the distribution is $var_l L \approx QN\lambda_l$. Both the mean and the variance are dimensionless as $\lambda_l$ is given in monomer units. (We have also derived exact expressions for the mean and the variance, but they are practically identical to the approximate ones for $QN, \lambda_l \gtrsim 2$, which is always the case.)

For the exponential depolymerizer gradient we find that

$$P_e(L) = \frac{1}{Z}e^{-\frac{\left(L - \lambda_e \log QN + \frac{1}{2}\right)^2}{2\lambda_e}}. \tag{9}$$

where to a good approximation the normalization constant is $Z \approx \sqrt{2\pi\lambda_e}$. In this case, the approximate mean length is, $\langle L\rangle_e \approx \lambda_e \log QN$ and the variance $var_e L = \lambda_e$; as above, the gradient decay length, $\lambda_e$, is expressed in units of monomers. In Fig 2 we compare our analytic results (Eqs (8) and (9)) to stochastic simulations of the chemical master Eq (7), and find almost perfect agreement.

In order to characterize the length fluctuations, we compute the noise for each model,

$$\mathrm{Noise} = \frac{\sqrt{\mathrm{Var}(L)}}{\langle L\rangle} \tag{10}$$

and its dependence on the model parameters. For the linear gradient model, using Eq (8) we find

$$\mathrm{Noise}_\ell \approx \frac{\sqrt{QN\lambda_\ell}}{(QN-1)\lambda_\ell} \approx (QN\lambda_\ell)^{-1/2} \approx \langle L\rangle_\ell^{-1/2}. \tag{11}$$

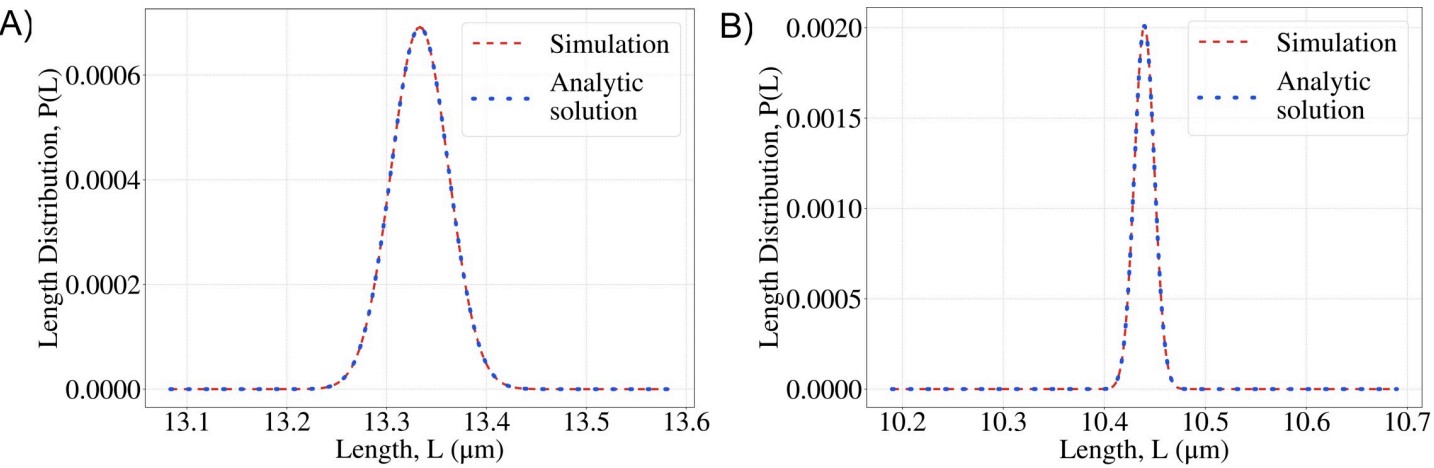

**Fig 2. Probability distributions of filament lengths A.** The linear gradient model and **B.** the exponential gradient model. Parameters: $N = 2 \times 10^6$, in both plots. For **A.** $QN = 5$, $\lambda_l = 6.66 \times 10^4$ (in monomer units). For **B.** $QN = 200$, $\lambda_e = 3.94 \times 10^4$ (in monomer units). (See Methods for estimates of the parameter values.) Simulation results are obtained from stochastic simulations of the chemical master Eq (7), and we compare them to analytical results, Eqs (8) and (9).

Therefore, as the average length increases, the noise decreases as the square root of the length, and the length of the filament becomes better defined.

Using Eq (9) for the length distribution in the exponential gradient model, we find

$$\text{Noise}_e \approx \frac{\sqrt{\lambda_e}}{\lambda_e \log QN - \frac{1}{2}} \approx \frac{1}{\sqrt{\lambda_e} \log QN} \tag{12}$$

and, once again, the model predicts that the length will become better defined as it increases.

If we compare our results for the length fluctuations, i.e., the widths of the distributions in Fig 2, to the experimentally measured length fluctuations [15] we observe that the theoretically predicted noise is $\approx 100$ times less than the observed noise.

While the computed variance from the model does not agree with experimental results, we do not believe that fluctuations in depolymerizer dynamics or the assembly and disassembly rates are the cause of these deviations due to aforementioned separation of timescales rendering these fluctuations ineffective. It should be noted that there are several other sources of noise presented in the experiments. For example, measurements of flagellar length are typically done using light microscopy and the diffraction limit of the microscopes can add significant amount of noise to the data.

**B. Limiting pool of monomers.** The two models investigated thus far describe stochastic assembly of filaments from a large pool of components and they both lead to well-defined length. Unlike the limited subunit-pool model [12,18] they can simultaneously regulate the lengths of multiple filaments. This is because the dynamics of filament assembly are independent of one another. Next, we examine the situation when the length dynamics of multiple filaments are coupled via a shared and limited pool of subunits to determine the effect of this coupling on length distributions.

For example, experiments on Chlamydomonas cells have shown that the assembly of its two flagella are not independent [15]. As described in the Introduction, when one flagellum is cut, the length of the intact flagellum decreases in response, while a new flagellum grows in place of the one that was removed. This observation supports the idea that the pool of tubulin used to assemble the flagella, is limited. Furthermore, it has been determined that in Chlamy

more than half of the tubulin in the cell is in its two flagella [15,31,32], which is also consistent with the idea of a limited tubulin pool.

The key question we address now is, how does the coupling of the filament-length dynamics via a shared and limited pool of monomers affect the mean and the variance of the length of each individual flagellum? We consider the master equation for two filaments:

$$
\begin{aligned}
\frac{dP(L_1, L_2, t)}{dt} &= k_+(N - L_1 - L_2 + 1)(P(L_1 - 1, L_2, t) + P(L_1, L_2 - 1, t)) \\
&\quad + \gamma(L_1 + 1)P(L_1 + 1, L_2, t) + \gamma(L_2 + 1)P(L_1, L_2 + 1, t) \\
&\quad - P(L_1, L_2, t)(2k_+(N - L_1 - L_2) + \gamma(L_1) + \gamma(L_2))
\end{aligned}
$$

$$
\gamma(L_i) = \begin{cases} k_- C_0 \left(1 + \dfrac{L_i}{\lambda_\ell}\right) & \text{Linear gradient} \\[2ex] k_- C_0 e^{L_i/\lambda_e} & \text{Exponential gradient} \end{cases}
$$

(13)

where $P(L_1, L_2, t)$ is the joint probability that the lengths of the two filaments are $L_{1,2}$ at time $t$ and $(N - L_1 - L_2)$ is the number of monomers left in the pool that are available for assembly. To obtain the steady state distribution of filament lengths, we set the left-hand side of Eq (13) equal to zero and solve for the probability $P(L_1, L_2)$, which is now time independent.

From the master equation, since the assembly and the disassembly process is reversible, the probability distribution satisfies the detailed balanced condition. It follows that the steady state probability distribution must satisfy the recursion relations,

$$
\gamma(L_1 + 1)P(L_1 + 1, L_2) = k_+(N - L_1 - L_2)P(L_1, L_2)
$$
$$
\gamma(L_2 + 1)P(L_1, L_2 + 1) = k_+(N - L_1 - L_2)P(L_1, L_2)
$$

and so, the solution of the master equation in steady state is given by,

$$
P(L_1, L_2) = \frac{1}{Z} \frac{k_+^{L_1 + L_2}}{f(L_1)f(L_2)(N - L_1 - L_2)!},
$$

(14)

where,

$$
f(L_i) = \begin{cases} \displaystyle\prod_{\ell=1}^{L_i} \gamma(\ell) & L_i > 0 \\[2ex] 1 & L_i = 0 \end{cases}
$$

From the joint probability distribution of lengths of the two filaments, we obtain the steady state distribution of lengths of a single filament, say $L_1$, by summing over all possible lengths of the other filament ($L_2$). In Fig 3A and 3B we plot the steady state probability distribution for a single filament, in the case of a linear and an exponential gradient of depolymerizers, using the same parameters as in Fig 2. To show the effect of a finite pool on the steady state length distribution, we compare the large pool and limiting pool versions of each model in Fig 3C and 3D. We see that the main effect of a finite monomer pool is that the mean lengths are smaller, while the noise practically stays the same. It changes from 0.007 to 0.006 for the linear gradient model, while for the exponential gradient model the difference is negligible. This conclusion does not change even when the number of monomers in the filaments is a much larger fraction of the monomer pool.

We explore the effect of the finite pool further in Fig 3E and 3F, where we plot the steady-state lengths when a finite monomer pool is taken into account and when it is not. Varying the length scale of the gradient λ, has the effect of changing the fraction of all the monomers in the

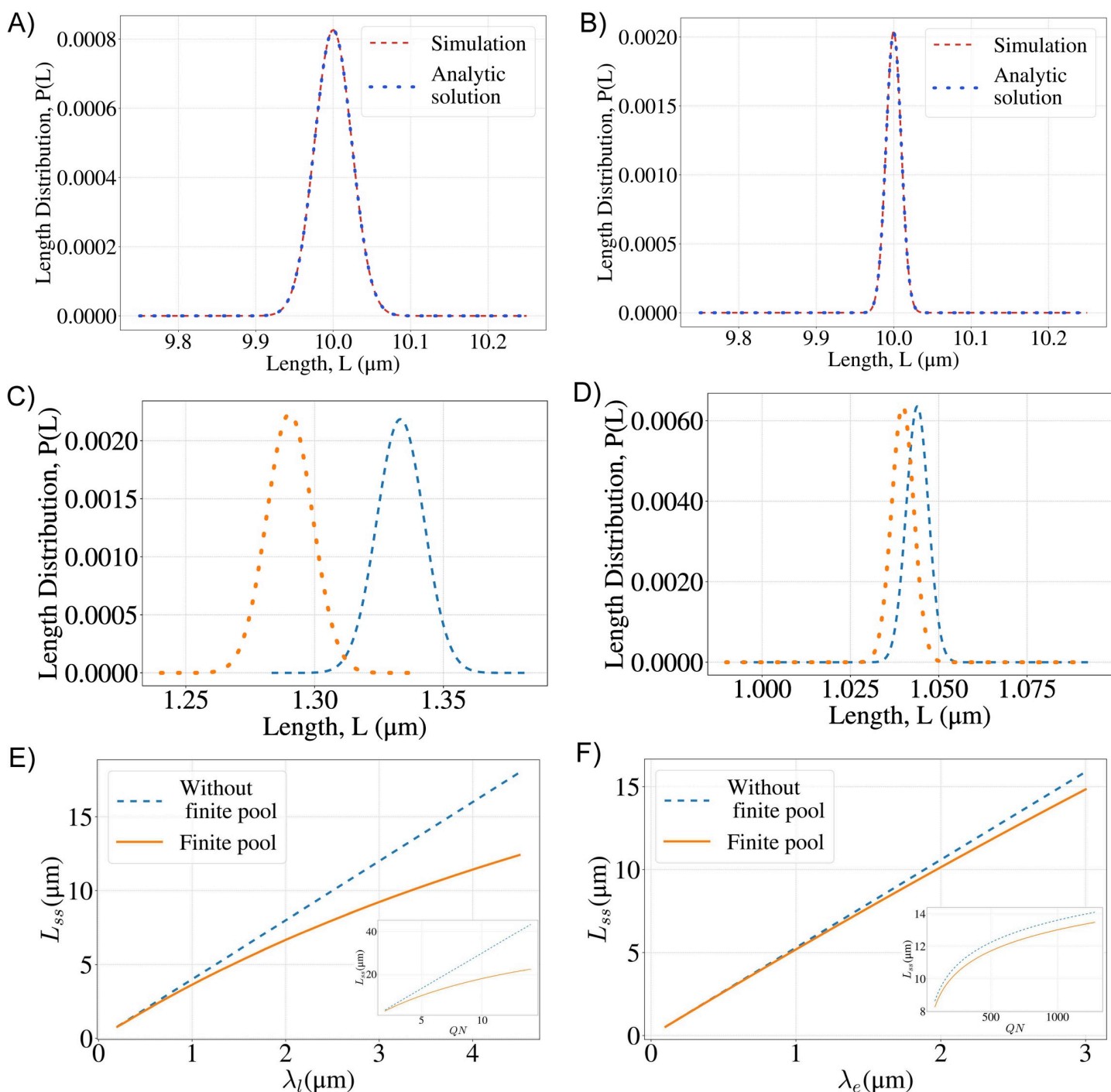

**Fig 3. Effect of a finite monomer pool on control of filament lengths. A,B** Marginal distributions of the filament lengths for the linear (A) and the exponential (B) gradient model with finite monomer pool. Simulation results are plotted together with Eqs (15) and (17). **C,D** Comparison of length distributions for the linear (C) and exponential (D) gradient models when the monomer pool is large versus infinite, when only a small fraction (~2%) of monomers is in filaments. In both plots, the orange dashed curve represents the finite pool model and the light blue dashed curve represents the large pool limit. **E, F.** The steady state lengths ($L_{ss}$) for the large monomer pool case (light blue) and the finite pool case (orange) for different values of the length scale of the gradient. Inset: steady state length for different values of $QN$ while keeping $N$ and $\lambda$ fixed.(E)-linear gradient model, (F)–exponential gradient model. Parameters: $N = 2 \times 10^6$. For **A**: $QN = 5$, $\lambda_l = 6.66 \times 10^4$ (in monomer units); **B**: $QN = 200$, $\lambda_e = 3.94 \times 10^4$ (in monomer units); **C**: $QN = 5$, $\lambda_l = 6.66 \times 10^3$ (in monomer units); **D**: $QN = 200$, $\lambda_e = 3.94 \times 10^3$ (in monomer units); **E**. $QN = 5$; Inset: $\lambda_l = 6.66 \times 10^4$ **F**: $QN = 200$. Inset: $\lambda_e = 3.94 \times 10^3$. (20,000 monomer units is one micron).

filaments, with smaller $\lambda$ corresponding to smaller fractions. As the fraction of monomers in the filaments decreases with decreasing $\lambda$, the difference between the predictions of the finite and large monomer pool case also decrease, as expected. For Chlamy, the steady state flagellum length is around 10 microns; for this value of $\lambda$, we find for the linear gradient model a difference of about 30% between the predicted steady state length from the finite and infinite pool cases, while for the exponential gradient model this difference is only 5%.

To further explore the consequences of the length distribution, Eq (14), we examine how the mean length and the noise of the distribution depend on the two parameters of the model, the dimensionless quotient of the assembly and disassembly rates ($QN$) and the length scale of the depolymerizing gradient ($\lambda$), while keeping the number of monomers ($N$) fixed.

For the linear gradient model Eq (14) can be further simplified to,

$$P_\ell(L_1, L_2) = \frac{1}{Z} \frac{(Q\lambda_\ell)^{L_1+L_2}}{(N - L_1 - L_2)!\,\Gamma(L_1 + \lambda_\ell + 1)\Gamma(L_2 + \lambda_\ell + 1)} \tag{15}$$

The mean and variance are not analytically tractable, but we can compute the steady state length from the system of equations $r(L_{\text{SS}}, L_{\text{SS}}) = \gamma_\ell(L_{\text{SS}})$, which yields

$$L_{\text{SS}} \approx \frac{N}{2} \frac{\lambda_\ell(QN - 1)}{QN\lambda_\ell + N/2}. \tag{16}$$

We plot this result in Fig 4A (which is practically indistinguishable from the mean length) as a function of $\lambda_\ell$, for different values of $QN$.

Next we numerically integrate Eq (15) to compute the variance of the length distribution for a single filament, and in Fig 4C we show how the noise of the distribution ($\sqrt{\text{Var}(L)}/\langle L \rangle$) depends on the parameters $\lambda_l$ and $QN$. We see that the noise is much less than one for a large range of parameters. This gives a measure of how well controlled the filament lengths are for different parameter values.

For the exponential gradient of depolymerizers, Eq (14) simplifies to

$$P_e(L_1, L_2) = \frac{1}{Z} \frac{e^{L_1 \ln Q - \frac{L_1(L_1+1)}{2\lambda_e}} e^{L_2 \ln Q - \frac{L_2(L_2+1)}{2\lambda_e}}}{(N - L_1 - L_2)!}. \tag{17}$$

In this case we cannot obtain an analytic formula for either the steady state length or the variance. The system of equations $r(L_{\text{SS}}, L_{\text{SS}}) = \gamma_e(L_{\text{SS}})$ does not have an analytic solution but is easily solved numerically to find approximate values for the steady state length, which we find to be an excellent approximation of the mean. As with the linear gradient model we find that the noise is much less than one for a range of parameters, Fig 4D.

## Discussion

Here we considered a simple model of length control of filaments undergoing rapid turnover of their constituent monomers, where the rate of depolymerization increases with filament length. Even though we only explore a two-filament system, our calculations can easily be extended to multiple filaments.

Length dependent depolymerization can be established in filamentous structures such as flagella and cilia, within which protein gradients are produced by a combination of directed motor transport and diffusion. If the protein being transported along the filament has a depolymerizing activity, then the accumulation of this protein at the filament tip (Fig 1) will lead to a length dependent depolymerization of the filament. Depending on the details of the transport, the amount of protein accumulated at the tip, and therefore its depolymerizing activity,

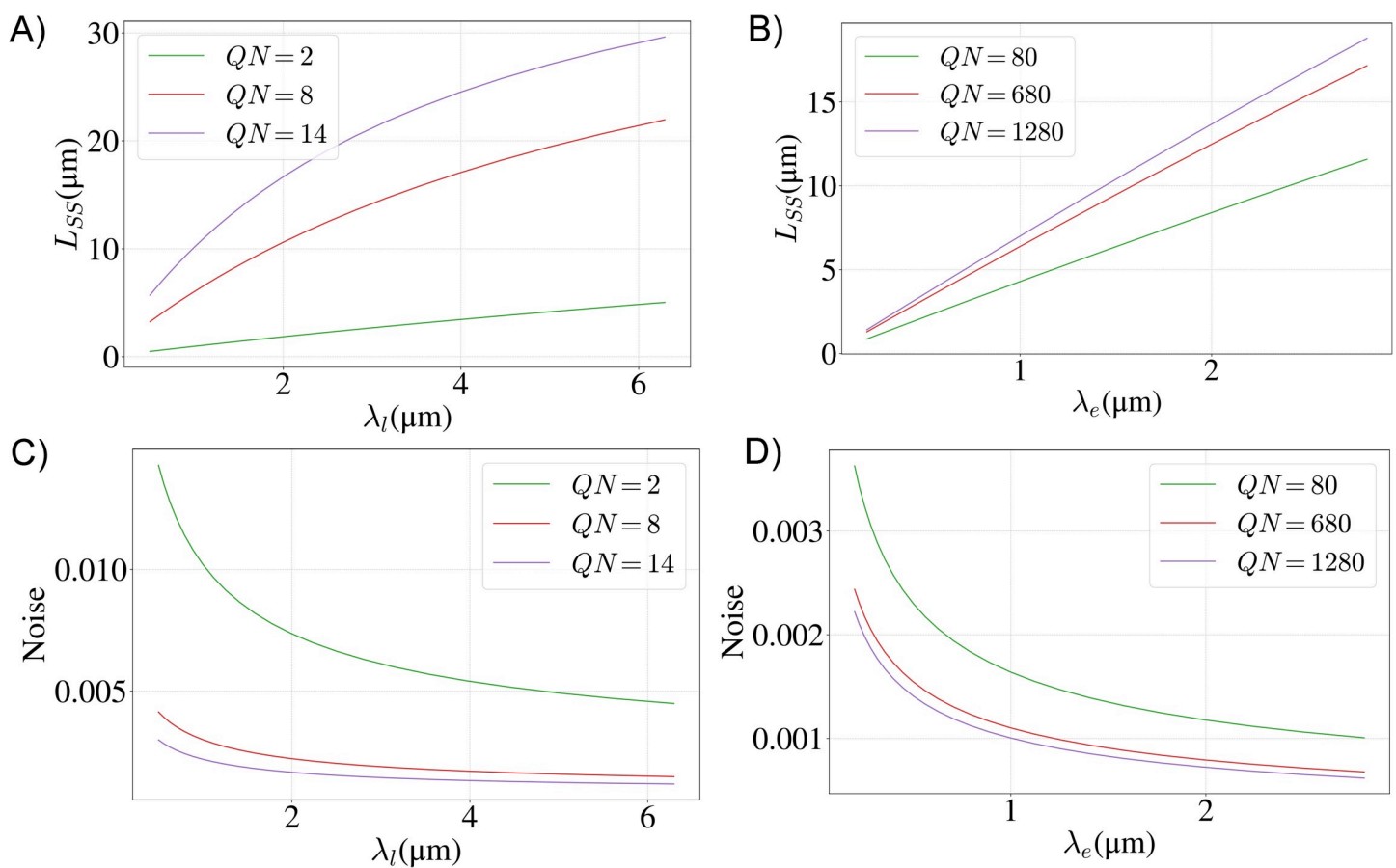

**Fig 4. Steady state length and noise for different parameter values.** Steady-state filament length for the linear (**A**) and exponential (**B**) gradient models, for different values of the dimensionless rate parameter $QN$ and gradient length scale $\lambda_{l,e}$. Noise of the length distribution for the linear (**C**) and exponential (**D**) gradient model for the same parameter values as in A and B. For all graphs the total number of monomers is $N = 2 \times 10^6$.

will either depend linearly or exponentially on the filament length [20,24]. Most importantly, in the presence of multiple filaments drawing from the same pool of depolymerizing proteins, the concentration at the tip of each filament will only depend on the length of that particular filament, thereby enabling simultaneous length control of multiple filaments [18]. This is in contrast to length control by a finite monomer pool in which case only the total length of all the filaments is precisely controlled, while each individual filament's length can vary wildly [12].

The key result of this paper are analytical expressions for the probability distributions of filament lengths when the dominant mechanism of length control is length-dependent depolymerization, due to either an exponential or a linear gradient of depolymerizing proteins. Notably, both types of protein gradients have been observed in flagella of single-cell organisms, Chlamydomonas and Giardia [19,25]. We derive analytical results for two scenarios: when the pool of filament subunits is limited (i.e., a sizable fraction of all monomers is in filamentous form in steady state), and when it is effectively infinite (i.e., the fraction of monomers in filaments is small). Estimates of model parameters based on experiments in Chlamydomonas yield steady-state lengths that are in agreements with observations. Based on the same parameters our estimates of noise are two orders of magnitude smaller than the observed length fluctuations, possibly pointing to a different source of noise not accounted for by our models.

Given that experiments that measure length fluctuations have been done using light micros-copy whose precision is diffraction limited, it could also be that the actual length fluctuations of the filaments are much smaller than those reported. This points to the need for more precise measurements of the length fluctuations of flagella as a way to discriminate between different mechanisms of length control.

A notable, qualitative difference between the linear and exponential gradient models is that in the latter case the steady state filament length is set by the length scale of the gradient and is only logarithmically dependent on the rates of polymerization and depolymerization (see Eq (6)). This makes the exponential gradient model more robust in controlling the length than the linear gradient model, for which the steady state length is more sensitive to model parame-ters, such as the total number of monomers and the concentration of depolymerizing proteins in the cell (see Eq (5)). Therefore, we are left with an interesting prediction of the exponential gradient mechanism, namely that orders of magnitude changes in the cytoplasmic concentra-tion of the depolymerizing protein will lead to much smaller changes in filament length (see Fig 4B). Future experiments that combine genetic and drug manipulations of cells in order to manipulate protein gradients within the flagellum, while at the same time measuring flagellar lengths, should provide stringent experimental tests of these predictions.

## Methods

### Estimation of model parameters

To make numerical estimates of the parameters that define the length distributions we make use of published data on flagella in Chlamy cells, whose steady state length is $L_{ss} \approx 10 \mu m$. Since each micron of a microtubule contains $\approx 2000$ tubulin dimers and each flagellum consist of roughly 10 microtubules, we estimate that there are around $2 \times 10^4$ tubulin dimers per micron of a flagellum. Therefore, a typical flagellum of length 10 microns contains $2 \times 10^5$ tubulin dimers. It has been estimated [31] that around 20% of the total amount of tubulin in the cell is used to make the flagella, so we estimate that the total number of tubulin dimers in the cell, $N = 2 \times 10^6$.

In ref. [25] a linear gradient of kinesin-2 proteins was measured along the length of the Chlamy flagella and this data provides an estimate of $\lambda_l \approx 3 \mu m$, for the characteristic length of the gradient. In the finite pool scenario, assuming a linear depolymerizing gradient model and a steady state length of 10 microns, gives $QN \approx 5$.

The dimensionless parameter $QN = \frac{k_+ N}{k_- C_0}$ is a ratio of the maximum polymerization rate (when all $N$ monomers are cytoplasmic) and the depolymerization rate at the cytoplasmic con-centration of the depolymerizing molecule ($C_0$). Based on experiments in which regrowth of a Chlamy flagellum is monitored after having been cut, we estimate the maximum polymeriza-tion rate to be a few microns a minute. Measurements of the depolymerization rate of MCAK [22] find that depolymerization of a few microns a minute is achieved at nanomolar concen-tration. Assuming that this is a good estimate for the depolymerization in the cell, we find $QN \approx 1$ for depolymerizer concentration ($C_0$) of a few nanomolar. If the actual cytoplasmic concentration of depolymerizer is in the picomolar range (which would correspond to tens of molecules of the depolymerizer in the Chlamy cell, which is tens of microns in diameter), we would end up with the estimate $QN \approx 1000$. Therefore, we take $1 < QN < 1000$ as a reasonable range of values for this dimensionless number. For example, in Fig 2, we have chosen $QN = 200$, which for the exponential gradient model with a finite pool of monomers and a steady state flagellum length of ten microns, gives an estimate $\lambda_e \approx 4 \times 10^4$ monomers or approx-imately 2 $\mu m$.

## Acknowledgments

We are grateful to Lishibanya Mohapatra, Thomas Fai, and Ariel Amir for useful discussions and comments on the manuscript.

## Author Contributions

**Conceptualization:** Arnab Datta, David Harbage, Jane Kondev.

**Data curation:** Arnab Datta, David Harbage, Jane Kondev.

**Formal analysis:** Arnab Datta, David Harbage, Jane Kondev.

**Funding acquisition:** Jane Kondev.

**Investigation:** Arnab Datta, David Harbage, Jane Kondev.

**Methodology:** Arnab Datta, David Harbage, Jane Kondev.

**Project administration:** Jane Kondev.

**Resources:** Jane Kondev.

**Software:** Arnab Datta, David Harbage.

**Supervision:** Jane Kondev.

**Validation:** Jane Kondev.

**Visualization:** Arnab Datta, David Harbage, Jane Kondev.

**Writing – original draft:** Arnab Datta, David Harbage, Jane Kondev.

**Writing – review & editing:** Arnab Datta, David Harbage, Jane Kondev.

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
