## [Decision Letter · Decision Letter 0]

30 Jul 2020

Dear Mr. Datta,

Thank you very much for submitting your manuscript "Control of filament length by a depolymerizing gradient" for consideration at PLOS Computational Biology.

As with all papers reviewed by the journal, your manuscript was reviewed by members of the editorial board and by several independent reviewers. In light of the reviews (below this email), we would like to invite the resubmission of a significantly-revised version that takes into account the reviewers' comments.

We cannot make any decision about publication until we have seen the revised manuscript and your response to the reviewers' comments. Your revised manuscript is also likely to be sent to reviewers for further evaluation.

Sincerely,

Martin Meier-Schellersheim

Associate Editor

PLOS Computational Biology

Daniel Beard

Deputy Editor

PLOS Computational Biology

Reviewer's Responses to Questions

**Comments to the Authors:**

Reviewer #1: In this article, the authors aim to establish a stochastic model of flagella growth and recession by introducing two separate but related methods of depolymerization that are dependent on the instantaneous length of the flagella. In these models, tubulin is transported to the free end of the flagella at a constant rate while a depolymerizing protein establishes a nonuniform concentration profile within the flagella that increases with distance from the base. The authors consider two types of concentration profiles: linear and exponential. In both cases, the probability distribution of flagella lengths is calculated as a function of the profile lengths scales and a defined dimensionless parameter. The statistical results, namely steady state length and relative noise, are compared to experimentally measured values. While steady state length is found to be in good agreement, calculations of relative noise are a full two orders of magnitude below the experimental values.

I find the foundations of this article to be strong and well motivated. Polymer dynamics is a rich and complex field, and the ubiquity of polymers throughout biology makes understanding how these dynamics function and are possibly regulated relevant for the study of most cells. The study performed here is narrowed to focus on the assembly and disassembly of flagella and other filamentous structures. The method used is a simple chemical master equation that expresses the length of such filaments as a stochastically evolving quantity. The resultant math is presented in a compact and easy to comprehend manner that highlights the major results of the study well. Importantly, the authors also present possible experiments that could be performed to test some of the predictions made by their models.

Despite these qualities, there are several areas of the article that should be addressed. In particular, I find two distinct major issues I feel need to be examined in some way within the article itself as well as a few other smaller remarks that I believe would make the paper stronger with minimal additional effort.

Major issues:

1) There are several different timescales inherent within the system the authors are attempting to model. While the authors note their approximation that the binding of depolymerizing protein to the tip of the filament is the rate limiting step when compared to the time needed to diffusively find the filament tip and the time to remove the terminal subunit, there are many more timescales that remain unmentioned. In the case of the linear gradient, there is the rate at which new depolymerizing protein attaches to motors at the proximal end, the time required for a motor to traverse the length of the filament, and the time required for a protein to diffuse back to the proximal end from the tip. The exponential gradient case also adds the rate at which diffusing protein in the filament rebind to already moving motors. These timescales need to be either included in the model or shown through experimentally measured values to be negligibly short. As an example, if the time to transport depolymerizing protein from the proximal to terminal end of the filament is of the same order as or longer than the time scale of the dynamics of the filament itself, then the concentration of depolymerizing protein at the tip will be a function not of the length of filament but of the length of the filament at a time in the past. This delayed reaction could have significant consequences on the steady state distribution of filament lengths. As such, the authors need to go back through their model, identify all possibly relevant time scales involved, and either include their effects in the model itself or show that they are realistically negligible.

2) Closely related to the issue of time scales is the issue of noise sources. In the chemical master equations presented in Eqs. 7 and 13, the only stochastic parameter is the filament length. However, concentration profiles of chemicals such as proteins are famously stochastic themselves. Fluctuations in the binding of filament subunits or depolymerizing protein to the motors can cause similar fluctuations in the assembly and disassembly rates of the filament respectively. Additionally, the depolymerizing proteins move through the filament diffusively once they detach from the motors. Diffusive fluctuations in the concentration can not only affect the concentration at the tip but also the rebinding rate to motors throughout the filament in the case of the exponential profile. As such, the assembly rate and depolymerizing protein concentration should themselves be treated as stochastic unless their inherent fluctuations can be shown to be either negligibly small or occur over timescales fast enough that they can be well approximated by their mean values. The addition of these noise sources could help explain the 100-fold discrepancy between the predicted and experimentally measured noise, though such a drastic shift in the predicted noise would itself be a concern unless thoroughly explored and understood.

Minor issues:

1) In the introduction the authors discuss experiments in which a flagellum of a Chlamydomonas cell is cut and the other shrinks in response until both reach the same length and regrow in unison. This is presented as a compelling argument for the validity of the shared monomer pool model, yet it is unclear as to whether the model presented here can reproduce this phenomenon. If it can, a visualization of such an occurrence using the model would strengthen the argument for the model’s overall validity. If not, this shortcoming should be addressed in the discussion.

2) In Eqs. 8 and 9, 1/Z is used as a normalized factor, but in Eqs. 14, 15, and 17 there is the factor A that is not explicitly stated to be a normalized factor, though it seems fairly obvious from context that that is its purpose. These notations should be uniform across the article with all equations using either 1/Z or A as the normalization factor.

3) When discussing Fig. 3E and 3F, the author note “As the fraction of monomers in the filaments decreases the difference between the predictions of the finite and large monomer pool case also decrease, as expected.” This should be shown in Fig. 3E and 3F by including multiple lines for finite monomer pools with varying fractions of the monomers being held within the filament.

4) In line 313 the authors state “which we find to be an excellent approximation (difference) of the mean.” I am unclear as to why the word “difference” is included in parentheses here as it seems to serve no purpose in the sentence.

Reviewer #2: In this paper the authors address the question how cells can control the length of multiple filamentous structures. In order for the structures to maintain a precise length, length-dependent assembly and/or disassembly rates are required. In their manuscript, the authors focus on length-dependent disassembly rates mediated by a depolymerase which is transported along the filamentous structure. Specifically, they assume that the depolymerase reduces the size of the filament with a typical rate once it reached the tip of a filament. Depending on the underlying dynamics of the depolymerase, distinct concentration profiles (of the depolymerase) emerge along the filament and thereby also different tip-occupations, which give rise to different effective disassembly rates. The authors model the depolymerase in an ad-hoc manner by considering different (typical) concentration profiles, namely a linear and an exponential concentration profile. They calculate the structure size distribution by solving the master equation for the underlying rate equation in the case of finite and infinite resources. They compare their theoretical results to experimental findings for biologically reasonable parameter estimates, and find good agreement for the mean structure size but the variance seems to significantly differ. A key prediction of their model is that the mean stationary state length of filaments in the linear gradient model depends linearly on the total amount of available assembly structures, whereas in the exponential gradient model the mean steady stationary length has a logarithmic dependence on the amount of available assembly structures.

Major Criticism

Since the authors derive the probability distribution of the underlying rate equations, their result for the variance is an important result of their analysis. However, they only briefly discuss the possible reasons for the clear deviations between the model and the experiments. From the perspective of the model, a natural explanation seems to us that the dynamics of the depolymerase is considered ad hoc via the concentration profiles. Actually, however, the depolymerase itself performs a stochastic movement. We would like to ask the authors to complement their discussion with stochastic simulation that explicitly considers the stochastic dynamics of the depolymerase.

The line of argument following line 146 is unclear to us. The authors use MCAK as an example for a linear relation between tip concentration, c(L), and effective depolymerization rate, ɣ(L). However, we would not consider MCAK to be the best example, because it actually performs a diffusive motion on the filament, which implies a flat concentration profile that does not correspond to the model assumptions (linear/exponential concentration profile) of the authors.

The directed transport of molecular motors with on/off kinetics is often modelled with the totally asymmetric simple exclusion process with Langmuir Kinetics. In the limit of low densities, such an analysis yields to a linear concentration profile as considered by the authors. However, the effective depolymerization rate would be determined by a flux balance equation at the tip, ε c(L)[1-c(l)]=γ(L), which implies a nonlinear dependence on the tip occupation. We would like to authors to comment on the influence of nonlinearity in the depolymerization rate on their results.

Minor Criticism

In line 269 they state that the steady state probability distribution must satisfy the relations given in line 271. We would suggest to add that the conditions given in line 271 correspond to the detailed balance condition which has to be fulfilled since the assembly process is time reversible. For a general stochastic process, we would not agree with the statement that the steady state probability distribution must fulfill the detailed-balance condition.

In Figure 3 E,F the biologically meaningful parameter range should be given so that one can judge whether the deviation between the finite and infinite pool model occurs for meaningful parameters.

**Have all data underlying the figures and results presented in the manuscript been provided?**

Reviewer #1: None

Reviewer #2: Yes

PLOS authors have the option to publish the peer review history of their article (what does this mean?). If published, this will include your full peer review and any attached files.

Reviewer #1: No

Reviewer #2: No
---

## [Decision Letter · Decision Letter 1]

12 Oct 2020

Dear Mr. Datta,

We are pleased to inform you that your manuscript 'Control of filament length by a depolymerizing gradient' has been provisionally accepted for publication in PLOS Computational Biology.

Best regards,

Martin Meier-Schellersheim

Associate Editor

PLOS Computational Biology

Daniel Beard

Deputy Editor

PLOS Computational Biology

Reviewer's Responses to Questions

**Comments to the Authors:**

Reviewer #1: The authors have sufficiently addressed all issues I found with the initial draft. My only recommended edits for this version are to rectify two simple typos:

1) Starting at the end of line 117, the authors state "In case when the loading of the depolymerizer". I believe this should read "In the case when the loading of the depolymerizer".

2) In line 156, the authors state "and knocks of monomers". I believe this should read "and knocks off monomers".

Reviewer #2: We thank the authors for the detailed answers to our questions and recommend the manuscript in its present form for publication.

**Have all data underlying the figures and results presented in the manuscript been provided?**

Reviewer #1: Yes

Reviewer #2: None

PLOS authors have the option to publish the peer review history of their article (what does this mean?). If published, this will include your full peer review and any attached files.

Reviewer #1: No

Reviewer #2: No

---

## [Editor Report · Acceptance letter]

23 Nov 2020

PCOMPBIOL-D-20-00970R1 

Control of filament length by a depolymerizing gradient

Dear Dr Datta,

I am pleased to inform you that your manuscript has been formally accepted for publication in PLOS Computational Biology. Your manuscript is now with our production department and you will be notified of the publication date in due course.

With kind regards,

Nicola Davies
